# Characterization of Genetic Variability of Common and Tartary Buckwheat Genotypes Using Microsatellite Markers

**DOI:** 10.3390/plants13152147

**Published:** 2024-08-02

**Authors:** Želmíra Balážová, Lucia Čišecká, Zdenka Gálová, Zuzana Hromadová, Milan Chňapek, Barbara Pipan, Vladimir Meglič

**Affiliations:** 1Institute of Biotechnology, Faculty of Biotechnology and Food Sciences, Slovak University of Agriculture in Nitra, Tr. A. Hlinku 2, 949 76 Nitra, Slovakia; lucia.mikolasova@gmail.com (L.Č.); zdenka.galova@uniag.sk (Z.G.); zuzanahromadova9@gmail.com (Z.H.); milan.chnapek@uniag.sk (M.C.); 2Crop Science Department, Agricultural Institute of Slovenia, Hacquetocva Ulica 17, SI-1000 Ljubljana, Slovenia; barbara.pipan@kis.si (B.P.); vladimir.meglic@kis.si (V.M.)

**Keywords:** SSR, *Fagopyrum*, UPGMA, PCoA, genetic diversity

## Abstract

Buckwheat is a highly nutritional pseudocereal with antioxidant potential. The aim of this study was to analyze the genetic variability of 21 varieties of common buckwheat (*Fagopyrum esculentum* Moench.) and 14 varieties of Tartary buckwheat (*Fagopyrum tataricum* Gaertn.) using microsatellite markers. By analyzing 21 SSR markers, an average of 11.6 alleles per locus were amplified and an average PIC value of 0.711 was determined. We determined the heterozygous status of the individuals and variability in the set using the SSR analysis on the basis of expected heterozygosity (He, 0.477), observed heterozygosity (Ho, 0.675), Shannon’s index (I, 0.820), and fixation indices (FST, FIS, FIT). Based on the SSR analyses, the lower level of expected heterozygosity in the analyzed set of Tartary buckwheat genotypes was observed compared to common buckwheat. With the help of a hierarchical cluster analysis using the UPGMA algorithm, Structure analysis, and PCoA analysis for the SSR markers, we divided the buckwheat varieties in the dendrogram into two main clusters according to the species. The AMOVA analysis showed that genetic variability between the individuals prevails in the analyzed set. The SSR technique proved to be a suitable tool for the determination of intra- and inter-varietal genetic variability and for analysis of diversity.

## 1. Introduction

Buckwheat (Polygonaceae family, *Fagopyrum* genus) is a commonly consumed crop in arid and cold regions of the world [1]. Buckwheat is a minor food crop that is classified as a pseudocereal [2]. The *Fagopyrum* genus consists of 23 species, of which the most important and the most widespread are 2 diploid species (2n = 2x = 16): common buckwheat (*Fagopyrum esculentum* Moench) and Tartary buckwheat (*Fagopyrum tataricum* Gaertn.) [1]. Common buckwheat is characterized by a short growth period and high resistance to environmental stress [3]. Tartary buckwheat is cultivated as a minor and popular crop in southwestern and northern China [4]. The short growing season and resistance to cold climate and high altitudes allow buckwheat to be grown in temperate Eurasia [5]. Buckwheat is considered to be a smart food crop of the future [2]. Its cultivation is widespread in the northern hemisphere, mainly in Asia, Europe, and North America. China is one of the world’s leading producers. Buckwheat most probably originated from southwest China where it is still cultivated in mountainous areas [6,7].

A buckwheat grain is rich in carbohydrates, fiber, lipids, minerals, but also antioxidants [1,8]. The amount of protein (12%) and fat (3%) in buckwheat is very similar to wheat. Globulin (43.3%) is the most abundant protein fraction, followed by glutelin (22.7%), albumin (18.2%), and prolamin (0.8%) [1,9]. Both common and Tartary buckwheat can be characterized by a high content of crude fiber and tannin [10]. A buckwheat grain contains proteins with a balanced composition of essential amino acids (methionine, tryptophan, lysine, histidine), and is rich in beneficial flavonoids, which also allow us to use buckwheat in the production of cosmetic, medical, and pharmaceutical preparations mainly thanks to its antioxidant, anti-inflammatory, antimutagenic, and anticarcinogenic properties [1,11,12]. Rutin stands out as the most important flavonoid; the highest concentration has been determined in the dry matter of Tartary buckwheat [13,14]. Consumption of buckwheat is beneficial for lowering cholesterol levels, controlling hypertension, diabetes, and preventing colon cancer. Buckwheat flour does not contain gluten, it is therefore suitable for patients suffering from celiac disease [1,8]. Despite its many benefits, buckwheat seeds also contain potent allergens that can induce anaphylactic reactions in sensitive patients. The buckwheat storage proteins, such as 13 S globulin and 2S albumin, have been reported as the major allergens of buckwheat [15].

In the last decade, great progress has been made in improving the properties of buckwheat and functional foods from buckwheat [10]. Due to its high protein content, antioxidants, and excellent low-cultivation properties, research on the genetic diversity of buckwheat is in high demand. Buckwheat is cultivated as a minor crop. But since it has many beneficial properties, breeders are developing new varieties with better yields and more desirable traits [5]. Molecular breeding of buckwheat has been held back due to the lack of genomic resources [16]. Information on genetic diversity is essential for the breeders to study the evolution of buckwheat [17]. Genetic diversity plays a key role in the survival of any species [7].

Genetic studies of buckwheat using a diverse gene pool are necessary to ensure the nutrition of the world’s population. The strategy should be to breed local specific varieties in different regions of the world [2].

Microsatellite markers (SSRs) are highly polymorphic, stable, and codominant [18]. These SSR markers have been used in the analyses of many crops, such as durum wheat [19], bread wheat [18,20,21], triticale [22,23], rice [24], maize [25,26], castor [27], and others, to assess their genetic diversity, phylogenetic relationships, and population structures [17]. Analyses of buckwheat using SSR markers have been performed by [1,17,28,29,30,31,32,33,34,35,36]. Microsatellite markers are a very effective tool for analyzing the genetic diversity and population structure of buckwheat [17]. Genetic mapping and construction of genetic maps was performed using the newly developed SSR markers, which proved to be highly efficient and provided useful information for the study of diversity and molecular breeding of the species in the *Fagopyrum* genus [11]. An assessment of the genetic diversity of 63 buckwheat genotypes was conducted using the SSR and ISSR markers by Sabreena et al. [34] and it showed that both marker techniques are highly efficient and informative for the detection of polymorphism in selected buckwheat genotypes. A genome-wide screening to develop the SSR markers associated with agronomic traits and rutin content for 97 genetic resources of Tartary buckwheat was evaluated by Hou et al. [32] in their study. The SSR markers were also used to analyze the morphological and molecular characterization of 112 varieties of Tartary buckwheat from 29 populations by Song et al. [7]. The SSR markers were also shown to be highly effective in determining the genetic purity of the Darja buckwheat cultivar [36]. The polymorphism among 52 genotypes of common buckwheat (*Fagopyrum esculentum*) was studied using 15 SSRs and furthered the evaluation of used SSRs for transferability to other buckwheat species [1]. The phenotypic and genetic diversity analysis of a global collection of the 2 cultivated buckwheat species *Fagopyrum esculentum* and *Fagopyrum tataricum* (190 and 51 accessions, respectively) using 37 agro-morphological traits and 24 SSR markers was performed by Pipan et al. [35].

The aim of our work was to (1) reveal the SSR-based polymorphism in a selected set of genotypes of common and Tartary buckwheat using 21 SSR markers, (2) assess the inter- and intra-species genetic variability, and also the variety-specific relationships in a selected set of genotypes of common and Tartary buckwheat, and at the same time to (3) prove the effectiveness of the used SSR markers for the analysis of the buckwheat genotypes in the identification and selection of genotypes for applicability in MAS (marker-assisted selection) breeding.

## 2. Results

### 2.1. Genetic Variability Based on the Occurrence of Alleles and Frequency of the SSR Markers Used

The 21 SSR primer pairs were used to analyze 35 genotypes of buckwheat of which 21 genotypes were common buckwheat (*Fagopyrum esculentum* Moench) and 14 genotypes were Tartary buckwheat (*Fagopyrum tataricum* Gaertn.). 

Altogether, 244 different alleles were obtained (Table 1). The number of alleles per locus ranged from 2 (TBP6) to 17 (SXAU089) with an average of 11.62 per locus. The number of effective alleles (Ne) was 2.326 on average. The length ranged from 126 bp to 478 bp. The average polymorphism of all loci was 82.720%.

The selected set of 21 SSR markers showed an average polymorphic information content (PIC) value of 0.711 (Table 1) that proved to be useful for differentiation of the analyzed genotypes of *Fagopyrum esculentum* Moench and *Fagopyrum tataricum* Gaertn. The PIC vales ranged from 0.007 (TBP6) to 0.851 (Fem1840). The Fem 1840 locus was determined as the most informative. For the Fem 1840 locus, the values of He 0.868, Ne 2.818 were determined. The marker index (MI) value that reflects its ability to differentiate the given genotypes using a selective marker was 0.711.

The observed heterozygosity (Ho) values ranged from 0.007 (TBP6 locus) to 0.986 (GB-FE-025, TBP5 loci), with an average value of 0.675. Compared to the observed heterozygosity, we obtained a lower average value of the expected heterozygosity (He) of 0.477. We did not notice a significant deviation from the Hardy–Weinberg equilibrium (HWE).

The Shannon index (I) values ranged from 0.011 (TBP 6 locus) to 1.093 (Ft3_572) in our analyzed set. The average value of I was 0.820 (Table 1), which indicates a relatively high diversity in the analyzed set of varieties of *Fagopyrum esculentum* Moench and *Fagopyrum tataricum* Gaertn.

### 2.2. Genetic Relationships in the Fagopyrum Species

Using the AMOVA analysis, 140 individuals from 35 populations revealed high molecular variation and significant genetic changes (*p* ≤ 0.01) across the populations and within the individuals (Table 2). A variation of 24% was found among the populations, with an estimated variance of 2.292, but no variance was found among the individuals. The highest variation of 76% was found within the individuals themselves, with an estimated variance of 7.086 (Table 2). An estimated variance of 9.378 was observed among and within the individuals of the populations studied. Based on the molecular variance observed, the fixation indices Fst of 0.294, Fis of −0.289, and Fit of 0.091 were observed among the populations based on their genetic content.

The results also showed that the selected species of common buckwheat and Tartary buckwheat are sufficiently diverse and the markers used are a suitable tool to differentiate the genotypes. This is evidenced by the lower demonstrated genetic variability within the species (46%) compared to the genetic variability between the species (56%), which we determined based on the AMOVA analysis.

The correlation indices based on allele frequency, locus by locus, include a set of parameters Fis, Fit, Fst (Table 2). The average inbreeding coefficient (Fis = −0.380) was negative, indicating that individuals from the set of genotypes are heterozygous.

The genetic diversity (Fst) values ranged from 0.122 (TBP6) to 0.498 (SXAU129) with an average value of 0.343 (Table 2). A moderate genetic diversity (Fst) value indicates moderate genetic differentiation. The mean value of excess heterozygotes (Fit) was 0.078.

### 2.3. Population Structure and Genetic Diversity Analysis between Fagopyrum Species

Using the Structure analysis, we were able to detect variability within a group of 35 genotypes using 21 SSR markers. Using all 35 genotypes, simulations were performed using Structure with K ranging from 1 to 10 and with 6 runs for K. The population groups in K were equal to two in this work (Figure 1 and Figure 2). As shown in the figure, the genotypes were divided into 2 groups: group I had 15 genotypes and group II had 20 genotypes. Based on the results obtained, we can assume the existence of genetic variability among the genotypes, which is an important aspect for studying the genetic diversity of the genotypes in question. The average determined genetic distance between the two clusters ranged from 0.464 (Cluster 1) to 0.688 (Cluster 2), with an average value of 0.576, which corresponded to the F statistics (Fst). The Fst values ranged from 0.125 (Cluster 1) to 0.453 (Cluster 2) with an average of 0.289.

A dendrogram was constructed using the UPGMA algorithm (Figure 3). The UPGMA dendrogram separated the genotypes of common and Tartary buckwheat into two main clusters (I, II). Twenty varieties of common buckwheat were separated in Cluster I. Cluster I was subsequently divided into two subclusters (Ia, Ib). In Subcluster Ia, the genotype Ballada (2) from Russia was separated from other common buckwheat genotypes. On the basis of Nei’s genetic distance coefficient, we evaluated the genotypes PI 481644 (22) and PI 481671 (23), both originating from Bhutan, as the genetically closest (0.038). The genotype Rana 60 (16) was separated from the common buckwheat genotypes and included in Cluster II where it was separated from the 14 genotypes of Tartary buckwheat. Cluster II was subsequently divided into two subgroups (IIa, IIb). A total of 12 out of 14 Tartary buckwheat genotypes clustered together, partly according to the country of origin. As the most genetically distant varieties within the interspecies analysis, we evaluated the common buckwheat variety Darina (5) from Slovenia and the Tartary buckwheat variety Weswod Ican (27) of unknown origin. For the mentioned varieties, we determined the genetic distance based on Nei’s coefficient of 1.668. The most diverse varieties could be used in breeding programs.

The PCoA analysis also showed that the genotypes of buckwheat were divided into two big groups and separated according to species. The Tartary buckwheat genotypes grouped in the red circle (quadrant 2, 3) and the genotypes of common buckwheat grouped in the blue circle (Figure 4). The common buckwheat variety Rana 60 (16) was assigned to the group of Tartary buckwheat genotypes (red circle).

### 2.4. Population Structure and Genetic Diversity Analysis within the Fagopyrum Species

Genetic diversity within the *Fagopyrum esculentum* Moench buckwheat species was also statistically evaluated. Four individuals from each variety were used in the analysis. The average value of the percentage representation of polymorphic loci was 92.29%. This indicates high genetic variability. The average determined value of the Shannon index (I) of 0.982 confirms the assumption of high genetic diversity in the analyzed set. The mean value of observed heterozygosity (Ho) (0.748) was slightly higher than the mean value of expected heterozygosity (He, 0.555). Based on the results it can be concluded that the set of buckwheat varieties is highly genetically diverse with a slight excess of heterozygous individuals (Table 3A).

By analyzing 21 varieties of common buckwheat, simulations were performed using the Structure 2.3.4 statistical software for K equal to 2 with 6 replicates. The buckwheat varieties were separated into two clusters (1, 2). Cluster 1 consists of 20 buckwheat varieties, and the Rana 60 (16) buckwheat variety was separated in Cluster 2 (Figure 5A).

The genetic distance between the two clusters varied from 0.683 (Cluster 2) to 0.712 (Cluster 1) with an average value of 0.698 (Table 3A), which corresponded to an F statistic (Fst) from 0.092 (Cluster 2) to 0.238 (Cluster 1) with an average of 0.165.

The results from the Structure analysis were also confirmed using the UPGMA algorithm. In the dendrogram constructed for the 21 varieties of *Fagopyrum esculentum* Moench (Figure 6A), the variety Rana 60 (16) was separated from the other varieties of common buckwheat genotypes and included into the group of Tartary buckwheat genotypes. The Ballada (2) variety was separated from other common buckwheat genotypes. Based on Nei’s coefficient (1.540), we identified the variety Emka (7) from Poland and Tohno Zairai (20) from Canada as the most genetically distant.

Genetic diversity was also statistically evaluated within the *Fagopyrum tataricum* Gaertn species in more detail using four individuals from each variety. The analyzed set of 14 varieties of Tartary buckwheat (*Fagopyrum tataricum* Gaertn.) showed a lower Shannon index value (0.578) compared to the value determined for the set of buckwheat varieties (0.982). We also recorded a lower average value of the percentage representation of polymorphic loci of 68.37%. We expected lower evaluated values of expected heterozygosity in the Tartary buckwheat varieties compared to common buckwheat due to the self-pollination of Tartary buckwheat. The observed that the heterozygosity value was higher than the expected heterozygosity value.

The analysis of 14 varieties of Tartary buckwheat was carried out using simulations using the Structure software for K equal to 2 with 6 repetitions. The number of clusters was two (1, 2). A total of 3 varieties of Tartary buckwheat originating from Bhutan (22, 23) and China (24) were grouped in Cluster I, while 11 varieties of Tartary buckwheat were grouped in Cluster II (Figure 5B). The average determined genetic distance between the clusters varied from 0.389 (Cluster 2) to 0.448 (Cluster 1) with an average value of 0.419 (Table 3B), which corresponded to the F statistic (Fst) from 0.062 (Cluster 1) to 0.291 (Cluster 2) with an average of 0.177.

Three varieties, PI 481644 (22), PI 481671 (23), and 903016 (24), which were separated into a separate cluster (Cluster I) based on the structural analysis, were also separated in the dendrogram constructed using the UPGMA algorithm (Figure 6B). Based on the Nei coefficient (0.367), we evaluated the variety named PI481644 (22) from Bhutan and the variety named Liuqiao-3 (32) from China as the most genetically distant. The most diverse varieties could be used in breeding programs.

## 3. Discussion

Research into the genetic diversity of buckwheat is currently needed to ensure the breeding of varieties with desirable traits with the aim to achieve the best possible traits in buckwheat, which is perceived as a functional crop of the future. The production of specific varieties of buckwheat with regard to its chemical composition and resistance to adverse environmental influences is a prerequisite to ensuring a sufficient quantity and quality of food.

The proper identification and characterization of genetic resources of plants is a key point in the breeding processes and for the maintenance of the genetic resources in the gene banks [37]. Molecular markers are the tool of choice to evaluate the genetic variability of genotypes and study their genetic relationships. Microsatellite markers are codominant, highly polymorphic, reproducible, and can be transferred among species. Each of the 21 SSR markers used was able to produce a different and polymorphic DNA. Altogether, 244 alleles were amplified. The results presented by authors [35] confirmed a high degree of polymorphism of the selected 24 SSR markers. Altogether, they detected 205 and 118 alleles in *F. esculentum* and *F. tataricum*, respectively. For the *Fagopyrum esculentum* accessions, the authors determined the maximum number of alleles (25) in the SXAU048 locus, while the lowest averaged allele values per locus were observed in the TBP6 locus (13.95). In our study, we also identified the TBP6 locus as the least polymorphic, determined by the lowest average number of alleles per locus (2). The authors [1] used 15 SSR markers to study polymorphism among 52 *Fagopyrum esculentum* genotypes. Of the 15 SSR markers, 14 were evaluated as polymorphic. They determined a total of 143 alleles in the set of genotypes of common buckwheat; most alleles were polymorphic with an average of 9 alleles per locus. By way of comparison, in our study we determined a slightly higher than average number of alleles per locus of 11.62 using 21 SSR markers. A lower average number of alleles (7.90 alleles per locus) was also obtained by Song et al. [17] and Grahić et al. [36] who determined an average of 7.1 alleles per locus. Lower average values of alleles were also reported by Ma et al. [31] who analyzed 41 populations of buckwheat using SSR markers and determined the average level of alleles of 5.90. Song et al. [7] also determined a slightly lower than average number of alleles (4.5) per locus.

The values of polymorphic information content (PIC) characterize the molecular markers in terms of variability and frequency of amplified alleles in the given populations, and thus represent the extent of the marker’s effectiveness in the determination of polymorphism. The PIC values calculated in our study ranged from 0.007 (TBP6) to 0.851 (Fem 1840) with an average PIC value of 0.711 per marker. The PIC values higher than 0.8 have been estimated using our data for the SXAU089, Ft2_2899, and Ft3_572 loci, and these results are in accordance with the results of [35]. The authors [35] identified the SXAU060 locus as the least polymorphic for *Fagopyrum tataricum*. Based on our data (PIC = 0.741; number of alleles = 12), we could conclude that the SXAU060 locus showed a high degree of polymorphism in the selected set of genotypes. The average PIC value (0.711) in our study was higher compared to the results presented by Bashir et al. [1] who achieved an average PIC value of 0.56. In the study of [1], the polymorphism information content (PIC) values for a set of common buckwheat genotypes ranged from 0.29 (primer GB-FE-054) to 0.92 (primer GB-FE-035). They evaluated GB-FE-035 (PIC = 0.92) as the most informative and highly effective SSR marker in the collection of common buckwheat genotypes, which was comparable to studies [17] and [34]. For the GB-FE-035 locus, we determined a lower average PIC value (0.679) in the set of common buckwheat and Tartary buckwheat genotypes. The GB-FE-035 locus marker also proved to be the most informative in analyzing the variability and population structure for 63 accessions of the *Fagopyrum* genus in the work of [34]. Using 15 SSR markers, Bashir et al. [1] also analyzed the transferability of selected SSR markers for a set of accessions of Tartary buckwheat (*Fagopyrum tataricum*). Out of 15 SSR primers, they determined 7 markers as evaluable. The total number of alleles was 136, with an average of 19 alleles per primer. The maximum number of alleles was determined for the Fem1322 (30) locus. The average PIC value per primer was 0.86. The Fem1322 locus was evaluated as the most polymorphic (PIC = 0.93). For the Fem 1322 locus, we determined a slightly lower PIC value (0.743) in the analyzed set of common buckwheat and Tartary buckwheat genotypes. The genetic diversity and agronomic traits of common buckwheat were analyzed by [28] using the AFLP and SSR markers. Five SSR markers showed a high degree of polymorphism. The authors determined an average PIC value of 0.84 and detected 203 alleles. The authors [28] reported the Fem1322 locus as the most informative. On the other hand, the authors determined the Fem1840 (PIC = 0.452) locus as the least informative. Based on our results, the Fem1840 locus was evaluated as the most informative and the most polymorphic (PIC = 0.851).

The authors of [28] determined the average expected heterozygosity (He) value as 0.819. In our study, we have achieved a comparatively lower value of He (0.477). Kishore et al. [38] analyzed the genetic diversity of 75 accessions of *Fagopyrum tataricum* using 15 SSR markers. The markers revealed high polymorphism (the average PIC = 0.93). The AMOVA showed that the genetic variability between 75 accessions of *Fagopyrum tataricum* was mainly between the populations (83.49%). On the contrary, in our set of the common buckwheat and Tartary buckwheat genotypes, the AMOVA analysis showed that the genetic variability in the analyzed set is most significantly influenced by the differences between the individuals (62%). Kishore et al. [38] reported the mean value of Shannon’s index (I) as 0.036, which is relatively lower than the I value determined in our study (0.820). It is important to note that the difference between the outcomes of the previously mentioned studies and our assessed data could be a consequence of using diverse genetic material or varying the amount of the SSR markers applied.

The dendrogram created on the basis of the SSR analyses in our study clearly separated the buckwheat species into two main clusters. Twenty common buckwheat varieties were separated into Cluster I. Fourteen Tartary buckwheat varieties were separated into Cluster II and the common buckwheat variety Rana 60 was assigned to this group as well. Based on the SSR and the other morphological characteristics of the seeds of the Rana 60 variety, such as seed shape, seed color, and seed size, we could include the Slovenian *Fagopyrum esculentum* Moench Rana 60 variety to the *Fagopyrum tataricum* Gaertn varieties. The variety Rana 60 also has the external morphological features of the plant, such as the shape of the leaves and thickness of the stem, which were comparable to the seeds and plants of the Tartary buckwheat varieties. The Structure analysis also confirmed our results of the UPGMA analysis. Different studies [1,35] showed that the SSR markers used are a suitable tool to differentiate between the *Fagopyrum esculentum* and *Fagopyrum tataricum* genotypes. Based on the SSR markers, the *Fagopyrum esculentum* and *Fagopyrum tataricum* species were also clearly separated into two groups in the work by Facho et al. [39], who analyzed the genetic material of buckwheat with 20 SSR markers. The hierarchical cluster analysis dendrogram divided the genotypes into three major clusters and found that the SSR and ISSR analyses were equally accurate in grouping the buckwheat genotypes according to their geographical origins [34]. In the study by [35], the SSR data also showed a clear genetic differentiation between common and Tartary buckwheat, confirming the cross incompatibility between the two species. However, within the species, the level of genetic diversity was not significantly different across the regions and the accessions were not geographically structured, indicating a weak genetic differentiation between the regions. The absence of such clustering suggests that a strong natural selection has been ongoing, resulting in continuous diversity.

## 4. Material and Methods

### 4.1. Biological Material

The genetic material analyzed in this study was obtained from the Gene Bank at the Research Institute of Plant Production in Piešťany, Slovakia and the Gene Bank at the Research Institute of Plant Production in Ruzyně, Prague, Czech Republic. The plant material analyzed (Table 4) consisted of 21 genotypes of common buckwheat (*Fagopyrum esculentum* Moench) and 14 genotypes of Tartary buckwheat (*Fagopyrum tataricum* Gaertn.). The genotypes of common buckwheat mostly originated from Austria (Bamby), Canada (Tohno Zairai), the Czech Republic (Pyra), France (La Harpe, St Jacut), Japan (Kasho-2), Poland (Emka, Hruszowska, Kora, Pulawska), Latvia (Aiva), Russia (Ballada, Bogatyr, Amurskaja FAG 29/79, Kazanska FAG 38/82), Slovakia (Špačinská 1), Slovenia (Darina, Darja, Rana 60, Siva), and the USA (Winsor Royal). The Tartary buckwheat genotypes were from Bhutan (PI 4816442, PI 4816712, 290, PI481661), China (Jianzui, Jinqiao-22, Liuqiao-32, Zhaoqiao-12), Mexico (PI 451723), Nepal (PI 427239), Pakistan (903016), and the USA (PI 476852, Sarasin a Ployes). The origin of the Weswod Ican genotype is unknown.

### 4.2. DNA Extraction of Plant Material

Genomic DNA was extracted from the young, healthy leaf tissue of four individual plants of each variety using the DNeasy Plant Pro Kit (Qiagen, Germantown, MD, USA). Homogenization was performed using the Retsch Tissue Lyser (Qiagen, USA). The DNA concentrations of each isolate were determined using a fluorimeter with the dsDNA Broad Range Assay Kit (Qubit 3.0; ThermoFisher Scientific, Waltham, MA, USA), with the DNA diluted to the final uniform concentration of 10 ng/µL.

### 4.3. Visualization and DNA Quantification

The quality of DNA was checked on the agarose gel with 0.5× Tris-borate-EDTA (TBE), stained with EtBr. The collected DNA samples were run on 1% agarose gels at 100 V at 35 min 5–6 µL of each sample and 3–5 µL of the loading buffer (XC+BB, Thermo Scientific, USA) was used alongside the GeneRuler 1 kb DNA Ladder (Thermo Scientific, USA).

### 4.4. PCR Amplification and Fragment Analysis

A total of 21 genus-specific SSR markers were used (Table 5). The PCR reactions were performed in a final volume of 11 µL that contained 1 µL of extracted DNA and the following reagents: 2 × PCR ToughMix (QuantaBio, Beverly, MA, USA), 0.1 μL of forward primer (10 μM; Sigma-Aldrich, St. Louis, MO, USA), 0.25 μL of reverse primer (10 μM; Sigma-Aldrich, USA), and 0.183 μL of 5′ fluorescently labeled universal primer (10 μM; 6-FAM, NED, HEX; Omega, Norwalk, CT, USA). The forward primer of each SSR had an added 18 bp tail sequence 5′-TGTAAAACGACGGCCAGT-3′ (M13(−21)), as described by Schuelke [40].

The amplification reactions were carried out using Verity (Thermo Fisher Scientific, USA) and SureCycler 8800 Thermal Cycler (Agilent Technologies, Santa Clara, CA, USA) with touchdown PCR conditions [42]. The individual steps in the PCR analyses were as follows: 94 °C for 4 min; fifteen cycles at 94 °C for 1 min, auto-decrease in temperature from 60 °C (62 °C) to 49.5 °C (51.5 °C) at 0.7 °C per cycle for 30 s, 72 °C for 1 min; followed by 23 cycles at 94 °C for 30 s, 53 °C for 30 s, 72 °C for 1 min; and the final extension for 5 min at 72 °C. The fragment analysis was performed on the 3130XL genetic analyzer (Applied Biosystems, Waltham, MA, USA). The allele lengths were determined by comparison with the GeneScan-500 ROX size standard (Applied Biosystems, USA) using the GeneMapper 4.0 software (Applied Biosystems, USA).

### 4.5. Data Analysis

Microsatellite-Toolkit [43] was used to calculate the observed number of alleles (No), allele frequency, expected heterozygosity (He), observed heterozygosity (Ho), and polymorphic information content (PIC). The GenAlEx 6.1 software [44] was used to determine the Shannon’s information index (I), fixation indices (Fis, Fst and Fit) and conduct the analysis of molecular variance (AMOVA) and principal coordinate analysis (PCoA). The UPGMA dendrogram according to Nei’s genetic distance was constructed by the Populations 1.2.28 software. The optimal value K for population structure analysis was calculated using the Structure harvester software 2.3.4 [45].

## 5. Conclusions

The selected SSR markers proved to be a suitable tool for the detection of polymorphism at the DNA level, which enabled an effective differentiation and characterization of the varieties of the analyzed set of common and Tartary buckwheat. The Rana 60 common buckwheat variety from Slovenia was classified as a Tartary buckwheat variety based on the results of the genomic SSR analyses. In the hierarchical cluster analysis, PCoA, and Structure analysis, the variety Rana 60 was assigned to the Tartary buckwheat varieties. The variety Rana 60 can be correctly classified with further characterization and analysis. Based on Nei’s coefficient, within the species of common buckwheat, we identified the varieties Emka (7) from Poland and Tohno Zairai (20) from Canada as the most genetically distant. Within the Tartary buckwheat species, we evaluated the variety named PI481644 (22) from Bhutan and the variety named Liuqiao-3 (32) from China as the most genetically distant. As the most genetically distant varieties between species, we evaluated the common buckwheat variety Darina (5) from Slovenia and the Tartary buckwheat variety Weswod Ican (27) of unknown origin. The most diverse varieties could be used in for the creation of varieties in the breeding process with improved agronomically important properties in MAS selection. The SSR markers with their codominant inheritance also allow us to distinguish the homozygotes from heterozygotes during crossing. Based on the self-pollination of Tartary buckwheat, we assumed a lower level of expected heterozygosity in the analyzed set of genotypes compared to common buckwheat, which was confirmed by our results. The varieties of common and Tartary buckwheat showed a significant genetic differentiation according to the SSR analysis, which confirms their cross-incompatibility. The analyses did not show a significant differentiation of the varieties based on their geographic origin, which may indicate intense natural selection and a high level of genetic diversity. The SSR markers used can be implemented in checking the uniformity of the newly created lines, as well as in the process of identifying the new varieties. They enable and speed up the selection process of varieties with improved key agronomic traits.

## Figures and Tables

**Figure 1 plants-13-02147-f001:**
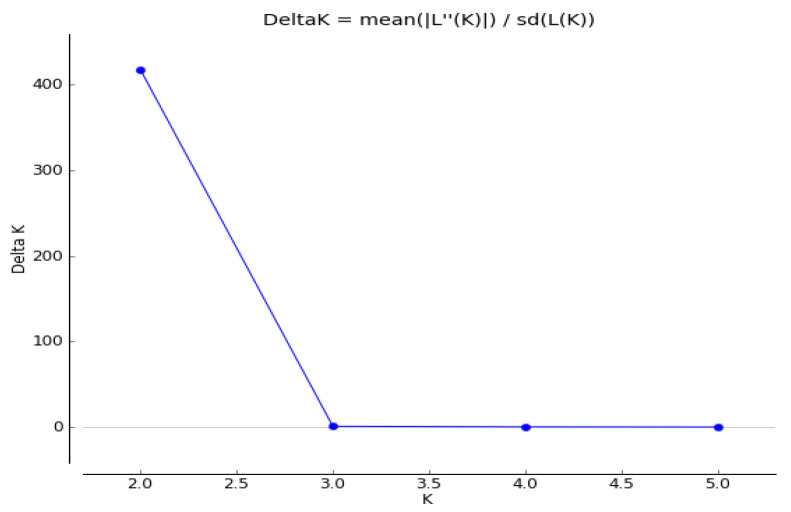
Structure plot of membership coefficients for all genotypes of buckwheat. The optimal number of clusters.

**Figure 2 plants-13-02147-f002:**
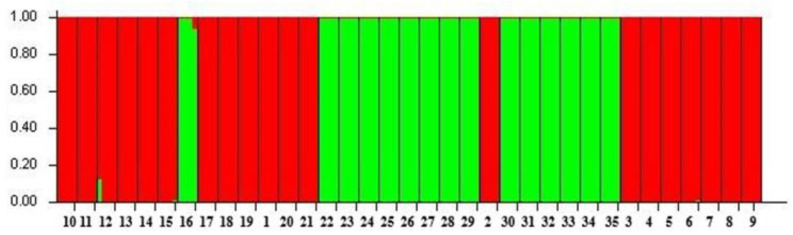
Population Structure analysis of *Fagopyrum esculentum* (red colour) and *Fagopyrum tataricum* (green colour) genotypes.

**Figure 3 plants-13-02147-f003:**
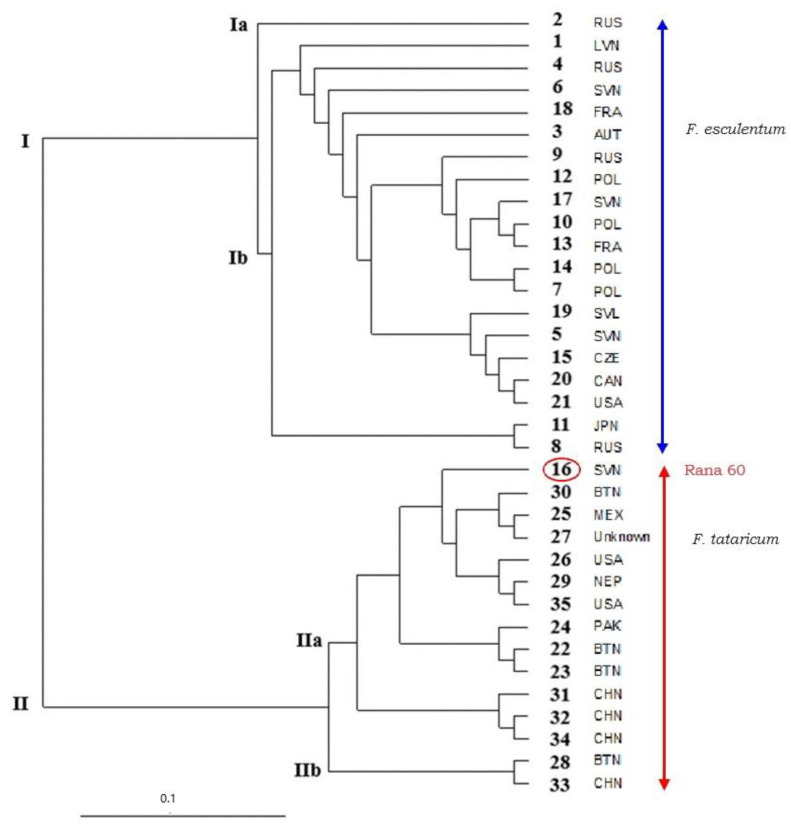
Dendrogram of 21 *Fagopyrum esculentum* and 14 *Fagopyrum tataricum* genotypes constructed using 21 SSR markers.

**Figure 4 plants-13-02147-f004:**
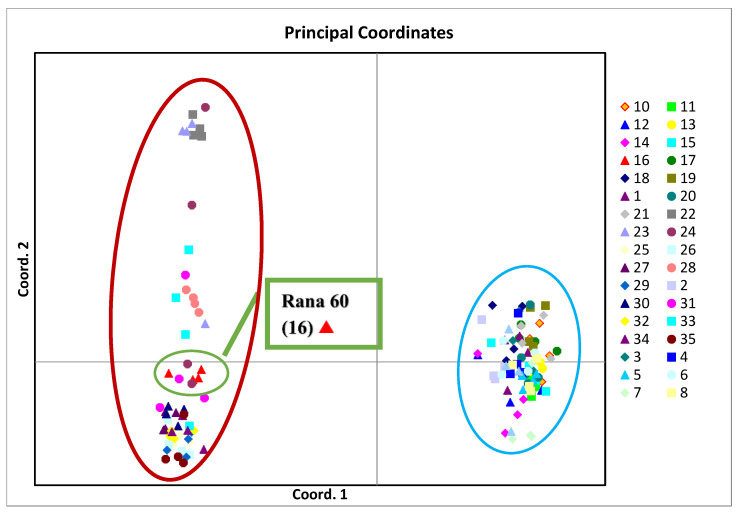
The PCoA plot of 21 *Fagopyrum esculentum* (blue circle) and 14 *Fagopyrum tataricum* (red circle) genotypes constructed using 21 SSR markers.

**Figure 5 plants-13-02147-f005:**
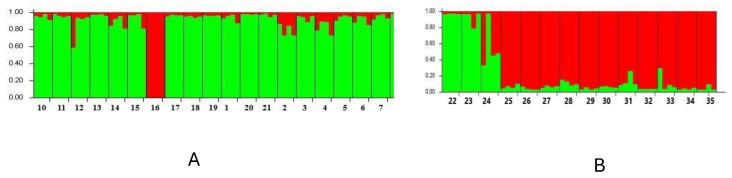
Structure analysis plots of the *Fagopyrum esculentum* Moench (**A**) and *Fagopyrum tataricum* Gaertn (**B**) varieties analyzed by 21 SSR markers.

**Figure 6 plants-13-02147-f006:**
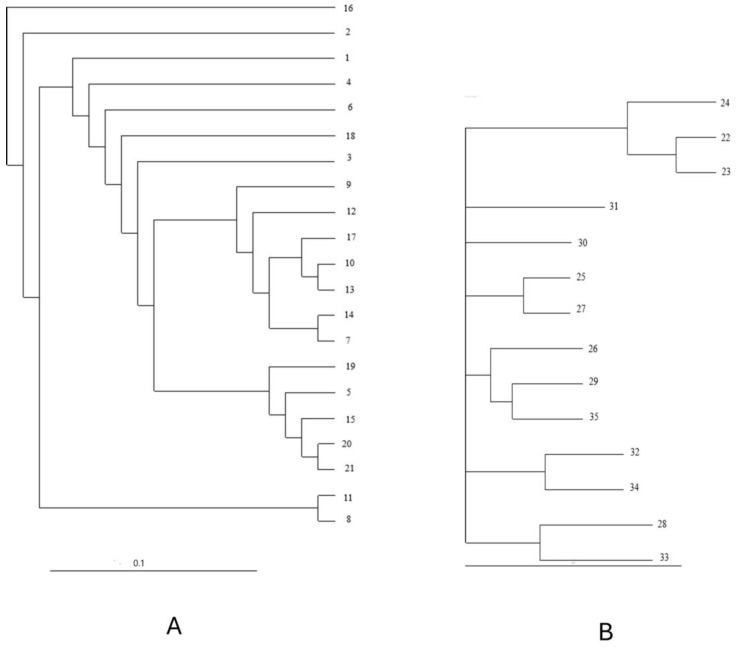
Dendrograms of 21 varieties of *Fagopyrum esculentum* Moench (**A**) and 14 varieties of *Fagopyrum tataricum* Gaertn. (**B**) constructed on the basis of 21 SSR markers.

**Table 1 plants-13-02147-t001:** Main statistical parameters for each microsatellite locus used.

Locus	Range of Allele Lengths (bp)	No	He	Ho	MI	PIC	Fst	Fis	Fit	I
Fem1303	198–354	13	0.549	0.614	0.742	0.742	0.416	−0.362	0.205	0.774
Fem1322	129–379	9	0.531	0.550	0.743	0.743	0.477	−0.369	0.284	0.716
Fem1407	173–225	12	0.600	0.886	0.782	0.782	0.263	−0.487	−0.097	1.041
Fem1840	170–296	13	0.525	0.736	0.851	0.851	0.299	−0.214	0.149	1.063
SXAU060	170–366	12	0.586	0.486	0.741	0.741	0.488	−0.234	0.368	0.681
SXAU089	190–266	17	0.610	0.471	0.803	0.803	0.413	0.020	0.425	0.857
SXAU129	276–370	9	0.394	0.421	0.673	0.673	0.498	−0.177	0.409	0.640
SXAU138	167–245	12	0.540	0.536	0.728	0.728	0.453	−0.307	0.285	0.792
Ft1_114	160–186	8	0.560	0.286	0.666	0.666	0.606	−0.022	0.597	0.482
Ft2_1743	152–327	16	0.549	0.714	0.802	0.802	0.355	−0.351	0.129	0.871
Ft3_572	143–478	10	0.555	0.907	0.829	0.829	0.266	−0.458	−0.071	1.093
Ft4_2725	232–322	13	0.531	0.921	0.812	0.812	0.292	−0.564	−0.107	0.973
Ft5_2899	140–310	12	0.583	0.936	0.818	0.818	0.250	−0.489	−0.117	1.083
Ft6_2849	202–332	6	0.311	0.543	0.427	0.427	0.260	−0.551	−0.148	0.527
Ft7_382	211–385	13	0.359	0.671	0.799	0.799	0.372	−0.312	0.176	0.917
Ft8_605	192–416	15	0.420	0.943	0.846	0.846	0.323	−0.617	−0.095	0.968
GB-FE-025	126–288	13	0.353	0.986	0.801	0.801	0.236	−0.570	−0.199	1.087
GB-FE-121	160–272	14	0.344	0.621	0.608	0.608	0.271	−0.367	0.003	0.818
GB-FE-035	172–390	14	0.348	0.950	0.679	0.679	0.179	−0.610	−0.322	0.983
TBP5	218–388	11	0.376	0.986	0.764	0.764	0.307	−0.789	−0.240	0.853
TBP6	205–211	2	0.369	0.007	0.007	0.007	0.122	−0.143	−0.004	0.011
Total		244								
Average		11.62	0.477	0.675	0.711	0.711	0.340	−0.380	0.078	0.820

Number of alleles (No); expected heterozygosity (He); observed heterozygosity (Ho); polymorphic information content (PIC); fixation indices (Fst, Fis, Fit); marker index (MI); Shannon’s information index (I).

**Table 2 plants-13-02147-t002:** Summary of the analysis of molecular variance (AMOVA).

Source	df	SS	MS	Est. Var.	%
Among Pops	34	756.539	22.251	2.292	24%
Among Indiv	105	410.750	3.912	0.000	0%
Within Indiv	140	992.000	7.086	7.086	76%
Total	279	2159.289		9.378	100%
F-Statistics	Value		P(rand ≥ data)		
Fst	0.294		0.010		
Fis	−0.289		1.000		
Fit	0.091		0.010		
Fst = AP/(WI + AI + AP) = AP/TOT					
Fis = AI/(WI + AI)					
Fit = (AI + AP)/(WI + AI + AP) = (AI + AP)/TOT					
Key: AP = Est. Var. Among Pops, AI = Est. Var. Among Individuals, WI = Est. Var. Within Individuals					

**Table 3 plants-13-02147-t003:** Heterozygosity and Fst values calculated for two clusters of *Fagopyrum esculentum* Moench (A) and *Fagopyrum tataricum* Gaertn (B).

(A)		
Cluster (K)	Genetic Distance	Fst for K
1	0.712	0.238
2	0.683	0.092
Average	0.698	0.165
(B)		
Cluster (K)	Genetic Distance	Fst for K
1	0.448	0.062
2	0.389	0.291
Average	0.419	0.177

**Table 4 plants-13-02147-t004:** List of analyzed buckwheat genotypes.

N	Accession Number	Accession Name	Genus	Species	Country of Origin
1	SVK001 Z50 00026	Aiva ^1^	*Fagopyrum*	*esculentum* Moench	LVA
2	SVK001 Z50 00004	Ballada ^1^	*Fagopyrum*	*esculentum* Moench	RUS
3	SVK001 Z50 00025	Bamby ^1^	*Fagopyrum*	*esculentum* Moench	AUT
4	SVK001 Z50 00003	Bogatyr ^1^	*Fagopyrum*	*esculentum* Moench	RUS
5	SVK001 Z50 00020	Darina ^1^	*Fagopyrum*	*esculentum* Moench	SVN
6	SVK001 Z50 00022	Darja ^1^	*Fagopyrum*	*esculentum* Moench	SVN
7	SVK001 Z50 00023	Emka ^1^	*Fagopyrum*	*esculentum* Moench	POL
8	SVK001 Z50 00015	Amurskaja ^1^ FAG 29/79	*Fagopyrum*	*esculentum* Moench	RUS
9	SVK001 Z50 00016	Kazanska ^1^ FAG 38/82	*Fagopyrum*	*esculentum* Moench	RUS
10	SVK001 Z50 00032	Hruszowska ^1^	*Fagopyrum*	*esculentum* Moench	POL
11	SVK001 Z50 00030	Kasho-2 ^1^	*Fagopyrum*	*esculentum* Moench	JPN
12	SVK001 Z50 00024	Kora ^1^	*Fagopyrum*	*esculentum* Moench	POL
13	SVK001 Z50 00005	La Harpe ^1^	*Fagopyrum*	*esculentum* Moench	FRA
14	SVK001 Z50 00035	Pulawska ^1^	*Fagopyrum*	*esculentum* Moench	POL
15	SVK001 Z50 00007	Pyra ^1^	*Fagopyrum*	*esculentum* Moench	CZE
16	SVK001 Z50 00021	Rana 60 ^1^	*Fagopyrum*	*esculentum* Moench	SVN
17	SVK001 Z50 00019	Siva ^1^	*Fagopyrum*	*esculentum* Moench	SVN
18	SVK001 Z50 00013	St Jacut ^1^	*Fagopyrum*	*esculentum* Moench	FRA
19	SVK001 Z50 00008	Spacinska 1 ^1^	*Fagopyrum*	*esculentum* Moench	SVK
20	SVK001 Z50 00028	Tohno Zairai ^1^	*Fagopyrum*	*esculentum* Moench	CAN
21	SVK001 Z50 00034	Winsor Royal ^1^	*Fagopyrum*	*esculentum* Moench	USA
22	01Z5100001	PI 481644 ^2^	*Fagopyrum*	*tataricum* (L.) Gaertn.	BTN
23	01Z5100009	PI 481671 ^2^	*Fagopyrum*	*tataricum* (L.) Gaertn.	BTN
24	01Z5100011	903016 ^2^	*Fagopyrum*	*tataricum* (L.) Gaertn.	PAK
25	01Z5100013	PI 451723 ^2^	*Fagopyrum*	*tataricum* (L.) Gaertn.	MEX
26	01Z5100014	PI 476852 ^2^	*Fagopyrum*	*tataricum* (L.) Gaertn.	USA
27	01Z5100017	Weswod Ican ^2^	*Fagopyrum*	*tataricum* (L.) Gaertn.	Unknown
28	01Z5100025	290 ^2^	*Fagopyrum*	*tataricum* (L.) Gaertn.	BTN
29	01Z5100030	PI 427239 ^2^	*Fagopyrum*	*tataricum* (L.) Gaertn.	NEP
30	01Z5100037	PI 481661 ^2^	*Fagopyrum*	*tataricum* (L.) Gaertn.	BTN
31	01Z5100041	Jianzui ^2^	*Fagopyrum*	*tataricum* (L.) Gaertn.	CHN
32	01Z5100042	Liuqiao-3 ^2^	*Fagopyrum*	*tataricum* (L.) Gaertn.	CHN
33	01Z5100044	Zhaoqiao-1 ^2^	*Fagopyrum*	*tataricum* (L.) Gaertn.	CHN
34	01Z5100046	Jinqiao-2 ^2^	*Fagopyrum*	*tataricum* (L.) Gaertn.	CHN
35	01Z5100050	Sarasin a Ployes ^2^	*Fagopyrum*	*tataricum* (L.) Gaertn.	USA

Sources of genetic material: ^1^ Gene Bank Piešťany, Slovakia, ^2^ Gene Bank Prague-Ruzyně, Czech Republic, Country of origin: SVN—Slovenia, SVK—Slovakia, POL—Poland, CZE—Czech Republic, AUT—Austria, BTN—Bhutan, CHN—China, RUS—Russia, LVA—Latvia, FRA—France, NEP—Nepal, USA—United States of America, PAK—Pakistan, MEX—Mexico, JPN—Japan.

**Table 5 plants-13-02147-t005:** Characteristics of microsatellite markers used in the genetic diversity analysis of buckwheat.

Marker	Sequence F	Sequence R	Repeat Motif	Reference
Fem1303	F: AGGAGACGGGAGAGAAGCAG	R: GGATGTTTGGGTGATTTCAG	(AG)_31_	[28]
Fem1322	F: AAGCATTCATTCATTCATTC	R: GAGTTTGTTGTGTTTGGAGG	(TC)_32_	[28]
Fem1407	F: GTGATGAGTAGTTGCCTCTG	R: CTTGGCTTAGACCTCTCGTA	(CT)_13_A(CT)_21_(CA)_9_	[28]
Fem1840	F: ACGACGAAGACAAATGAGGA	R: ATATGGACGGCCTGGATTAT	(GA)_8_	[28]
SXAU060	F: TCCCAATAGCCAATAGTACATG	R: GACCTAATTAACCGTTAGCACA	(AAT)_10_	[32]
SXAU089	F: CAAAAGAAAAGTGCCGAAGT	R: TTATGTCACCGCCATTGTT	(CCA)_10_	[32]
SXAU129	F: CTCAAAGGATGCCATTGTAAC	R: GACTTTGAGAACGCCTTGAC	(TA)_11_	[32]
SXAU138	F: CACCTGCTACAATACTCTCA	R: GCTTAATCAACAGTAGGCAC	(AGT)_10_	[32]
Ft1_114	F: CAACAGCATCTTTCCCTTCA	R: CCATAAACACAGCAACAGCC	(T)_24_	[11]
Ft2_1743	F: ACCACTGACAATAAGGGGGA	R: CAAAAGGTTGATGTGGATGG	(AT)_37_	[11]
Ft3_572	F: CATCACCCCTCTCAAGACCT	R: AGAATCCTACCCCGTCCTTT	(TA)_14_	[11]
Ft4_2725	F: TAGCGATTTGAAGGGGACTT	R: CGTAACAATGGTCGTTACTCG	(AT)_10_	[11]
Ft5_2899	F: AAGCTTCCTTCCATGACCAC	R: GTTTCTTGTGTGGACCGTTG	(TA)_11_	[11]
Ft6_2849	F: ACAATTCATCAAGCGACTCC	R: CTTTGCCGAATGTAGGGAAT	(AAAAC)_4_	[11]
Ft7_382	F: TGGTCTCTAAAACGGACCGA	R: TCGGAACCGGATTCTCTTAC	(AT)_27_	[11]
Ft8_605	F: AACGAGGGTACTAACCGGAA	R: CCCCAGCTGTAAAACAATCA	(ATA)_19_	[11]
GB-FE-025	F: CAGATGTTACCCGAGGCA	R: ACCCATATGTCACGAGCG	(ACCTCC)_6_	[31]
GB-FE-121	F: TCCATACCAAGCAGGTGG	R: GTGCCTGATGAGGTTCCA	(AGG)_6_	[31]
GB-FE-035	F: TGCAATGACTTGGAGGAGA	R: ACCACCATTCAACAAGCG	(GAK)_6_ (GAT)_3_ (GAT)_2_ K(G/T)	[31]
TBP5	F: GGGGATTGATCGAGAAAG	R: CCGAAGAGTAACTAGGAC	-	[41]
TBP6	F: CGGCTAATAAGTCGTTTC	R: GGATCATAGGTCGTGAAT	-	[41]

## Data Availability

The original contributions presented in the study are included in the article/Appendix A, further inquiries can be directed to the corresponding author.

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
