# Peer review of "Characterization of Genetic Variability of Common and Tartary Buckwheat Genotypes Using Microsatellite Markers"

_plants, 2024, doi:10.3390/plants13152147_

Round 1

Reviewer 1 Report

Comments and Suggestions for Authors

Comments and Suggestions for Authors

Dear Author,

It is my pleasure to review the manuscript entitled “Characterization of genetic variability of common and Tartary buckwheat genotypes using microsatellite markers” a research article submitted to MDPI Journal, Plants. Authors of this manuscript studied genetic diversity, variability, and cluster analysis of 21 varieties of buckwheat and 14 varieties of Tartary buckwheat accessions using 21 SSR markers and several agronomic traits. They have performed series of genetic diversity related morpho-physiology study and molecular markers to select genotypes to be further used in breeding program for sustainable development. Overall, the experiments, they performed, are well and the results are convincing. However, I have some suggestions, which might improve the manuscript to make important to the wider readers.

-Grammar need to be improved. I suggest a careful revision by an expert.

-There are some issues that I have already mentioned in the pdf file of the submitted manuscript to be reconsidered

1. Introduction

-Introduction should be more focused on the research topic. Huge vague and not related description. A large amount of research on genetic diversity and current molecular markers on wheat and related plants as are already available in the literature. The present work does not add much new insights of wheat diversity. Rationale of the study should be elucidated clearly. Why further this research is necessary, this should be stated very strongly.

-Last sentence should be more specific and informative of results obtained

-The aim of the study should be clearly depicted and fulfilled by the data presented and the conclusions using appropriate references.

- Some references are quite old, that could be replaced by more recent ones

4. Materials and Methods

-For SSR analysis, only 21 markers were used which is quite low number for making right prediction

-Make clear, how many for data collection and which for DNA sampling

-Reorder work style. First grown and then sample collection

-Describe plant cultivation condition. Mention about temperature, field condition, soil texture and organic/inorganic content of the soil of culture environment

-You do not have any morphology study. How the plants were phenotypically differed? Details on phenotyping should be given if you have?

-The selection strategy of these markers is confusing. Are these linked to some traits/ random markers were taken?

-Only 1.4% gels were used for electrophoresis which limits the detection of many alleles.

-It is necessary to provide PCR gel pictures at least in the supplementary file for few markers with all genotypes

2. Results

-Figure indication in the text should be based on fig. position. Fig. 1 should be first then next. Not followed here. Please carefully check it. In the text, describe first, then place figure.

-Same for table. Follow chronology

- 2.3. Population structure and genetic diversity analysis within the Fagopyrum species; it is like mostly repetition

-format all tables with good presentation. No inner lines

-Table 3. Are those SSR markers randomly selected? Do you have any reference? Chromosome number should be chronological

-Table 5 and 6 should place earlier

3. Discussion

- The authors should discuss more thoroughly the results obtained regarding further use of the genotypes in breeding strategies for yield improvement

Comments on the Quality of English Language

Author Response

Dear reviewer,

Thank you for your comments of our article, here are our answers.

-Grammar need to be improved. I suggest a careful revision by an expert.

The English grammar was already revised by an English expert, I am adding the evidence of it in the attached word document.

-There are some issues that I have already mentioned in the pdf file of the submitted manuscript to be reconsidered

  1. Introduction

-Introduction should be more focused on the research topic. Huge vague and not related description. A large amount of research on genetic diversity and current molecular markers on wheat and related plants as are already available in the literature. The present work does not add much new insights of wheat diversity. Rationale of the study should be elucidated clearly. Why further this research is necessary, this should be stated very strongly.

Introduction was corrected.

The article deals with the pseudocereal buckwheat not cereals like wheat, thats´way we do not describe the genetic diversity research concerning cereals like wheat.

-Last sentence should be more specific and informative of results obtained

Last sentence was changed.

-The aim of the study should be clearly depicted and fulfilled by the data presented and the conclusions using appropriate references.

The aim was changed.

- Some references are quite old, that could be replaced by more recent ones

Some references were replaced by more recent ones.

  1. Materials and Methods

-For SSR analysis, only 21 markers were used which is quite low number for making right prediction.

In the article was confirmed that the number of markers (21) was sufficient to distinguish between common and tartary buckwheat genotypes and we could even identify the genotype of common buckwheat Rana 60 (16), which was primary included into the Fagopyrum esculentum, and suggest the assignment to another species - tartary buckwheat. Also the other researchers used similar number of molecular markers and declared that the similar number markers was sufficient to differentiate genotypes (Pipan et al., 2023 – 24 SSR markers https://doi.org/10.3390/plants12183321,  Bashir et al., 2021 – 15 SSR markers https://doi.org/10.1007/s13237-020-00319-y) and others.

-Make clear, how many for data collection and which for DNA sampling

Corrected.

-Reorder work style. First grown and then sample collection

Corrected.

-Describe plant cultivation condition. Mention about temperature, field condition, soil texture and organic/inorganic content of the soil of culture environment

Corrected.

-You do not have any morphology study. How the plants were phenotypically differed? Details on phenotyping should be given if you have?

The phenotypic differentiation was not the aim of the study, we don’t have this information.

-The selection strategy of these markers is confusing. Are these linked to some traits/ random markers were taken?

The SSR markers were chosen based on preliminary research made by Crop Science Department, Agricultural Institute of Slovenia. SSR markers were chosen the most polymorphic ones and were chosen also based on transferability between species and from different authors (Table 5). 

-Only 1.4% gels were used for electrophoresis which limits the detection of many alleles.

Corrected.

-It is necessary to provide PCR gel pictures at least in the supplementary file for few markers with all genotypes

PCR visualisation in the gel was done only as the control of the presence of amplification products, the gels were not documented. After PCR reaction the fragment analysis was performed on the 3130XL genetic analyser (Applied Biosystems, USA).

  1. Results

-Figure indication in the text should be based on fig. position. Fig. 1 should be first then next. Not followed here. Please carefully check it. In the text, describe first, then place figure.

Corrected.

-Same for table. Follow chronology

Corrected.

- 2.3. Population structure and genetic diversity analysis within the Fagopyrum species; it is like mostly repetition

Population structure and genetic diversity analysis within the Fagopyrum species in the part 2.3. more deeply characterize the species individually using UPGMA and Structure analysis, it not the repetition of the research.

-format all tables with good presentation. No inner lines

Corrected.

-Table 3. Are those SSR markers randomly selected? Do you have any reference? Chromosome number should be chronological

The SSR markers were chosen based on preliminary research made by Crop Science Department, Agricultural Institute of Slovenia and were chosen the most polymorphic, based on transferability between species and from different authors (Table 5).  The SSR markers are written alphabetically by name.

-Table 5 and 6 should place earlier

Table 5 and 6 (changed to table 4 a 5) are placed at the end of the article, they belong to the section Material and Method, they can´t be placed earlier, because the Material and Method are at the end of the article.

  1. Discussion

- The authors should discuss more thoroughly the results obtained regarding further use of the genotypes in breeding strategies for yield improvement.

Corrected, changed.

Reviewer 2 Report

Comments and Suggestions for Authors

Manuscript "Characterization of genetic variability of common and Tartary buckwheat genotypes using microsatellite markers" is very interesting.

General comments:
Authors analyzed the genetic variability and assessed the inter- and intra species, and also the variety-specific relationships in a selected set of genotypes of common and Tartary buckwheat using 21 SSR markers.

The results presented are comprehensive and completely descriptive of the study population. The authors have characterized the studied genotypes by multiple methods, which are correct.

Detailed comments:
Figure 5: Provide percentages for each component.
Table 3: Replace commas with periods.
Table 4: Replace commas with periods.

Paper needs minor revision.

Author Response

Detailed comments of the Reviewer 2:

Figure 5: Provide percentages for each component.

This comment was not clear, because the Fig. 5 (Plot analysis) doesnt contain any percentage. We dont understand what the reviewer means.

Table 3: Replace commas with periods.

Corrected.

Table 4: Replace commas with periods.

Corrected.

Round 2

Reviewer 1 Report

Comments and Suggestions for Authors

Even some points are not corrected, however, the article has been improved substantially